# Distinct combinations of variant ionotropic glutamate receptors mediate thermosensation and hygrosensation in *Drosophila*

Zachary A Knecht[1†], Ana F Silbering[2†], Lina Ni[1†], Mason Klein[3,4†], Gonzalo Budelli[1], Rati Bell[2], Liliane Abuin[2], Anggie J Ferrer[4], Aravinthan DT Samuel[3*], Richard Benton[2*], Paul A Garrity[1*]

[1]Department of Biology, National Center for Behavioral Genomics and Volen Center for Complex Systems, Brandeis University, Waltham, United States; [2]Center for Integrative Genomics, Faculty of Biology and Medicine, University of Lausanne, Lausanne, Switzerland; [3]Department of Physics and Center for Brain Science, Harvard University, Cambridge, United States; [4]Department of Physics, University of Miami, Coral Gables, United States

**\*For correspondence:**
aravisamuel@me.com (ADTS);
Richard.Benton@unil.ch (RBen);
pgarrity@brandeis.edu (PAG)

[†]These authors contributed equally to this work

**Competing interests:** The authors declare that no competing interests exist.

**Abstract** Ionotropic Receptors (IRs) are a large subfamily of variant ionotropic glutamate receptors present across Protostomia. While these receptors are most extensively studied for their roles in chemosensory detection, recent work has implicated two family members, IR21a and IR25a, in thermosensation in *Drosophila*. Here we characterize one of the most evolutionarily deeply conserved receptors, IR93a, and show that it is co-expressed and functions with IR21a and IR25a to mediate physiological and behavioral responses to cool temperatures. IR93a is also co-expressed with IR25a and a distinct receptor, IR40a, in a discrete population of sensory neurons in the sacculus, a multi-chambered pocket within the antenna. We demonstrate that this combination of receptors is required for neuronal responses to dry air and behavioral discrimination of humidity differences. Our results identify IR93a as a common component of molecularly and cellularly distinct IR pathways important for thermosensation and hygrosensation in insects.

## Introduction

Ionotropic Receptors (IRs) are a large subfamily of ionotropic glutamate receptors (iGluRs) that appear to have evolved in the last common protostome ancestor (*Benton et al., 2009*; *Croset et al., 2010*; *Rytz et al., 2013*). In contrast to the critical role of iGluRs in synaptic communication, IRs have diverse roles in chemosensory detection (*Koh et al., 2014*; *Rytz et al., 2013*). The best-defined functions of IRs are in olfaction, where they mediate sensory neuron responses to diverse chemicals, including many acids and amines (*Rytz et al., 2013*; *Silbering et al., 2011*). Most IRs are thought to form heteromeric ligand-gated ion channels, in which broadly expressed co-receptor subunits (e.g., IR8a, IR25a and IR76b) combine with more selectively expressed IR subunits that confer stimulus specificity (*Abuin et al., 2011*; *Rytz et al., 2013*). Many of these IRs are highly conserved in insects, indicating that they define sensory pathways common to a wide range of species (*Croset et al., 2010*; *Rytz et al., 2013*).

Although most conserved IRs have been assigned chemosensory roles, we recently reported that one of these receptors, IR21a, mediates cool sensing (together with IR25a) in a population of neurons in the *Drosophila melanogaster* larva, the dorsal organ cool cells (DOCCs) (*Ni et al., 2016*).

This finding raised the possibility that other IRs serve non-chemosensory functions. In this work we characterize one of these 'orphan' receptors, IR93a, which has orthologs across arthropods (*Corey et al., 2013*; *Groh-Lunow et al., 2014*; *Rytz et al., 2013*). RNA expression analysis in several insects and crustaceans indicates that this receptor gene is transcribed in peripheral sensory organs (*Benton et al., 2009*; *Corey et al., 2013*; *Groh-Lunow et al., 2014*; *Rytz et al., 2013*), but its role(s) are unknown. Using *Drosophila* as a model, we find that IR93a acts with different combinations of IRs in distinct populations of neurons to mediate physiological and behavioral responses to both thermosensory and hygrosensory cues.

## Results

### IR93a is expressed in larval thermosensory neurons and is essential for cool avoidance

To investigate the expression and function of IR93a, we generated antibodies against a C-terminal peptide sequence of this receptor and obtained two *Ir93a* mutant alleles: *Ir93a*$^{MI05555}$, which contains a transposon insertion in the fifth coding exon, and *Ir93a*$^{122}$, which we generated using CRISPR/Cas9 to delete 22 bases within the sequence encoding the first transmembrane domain (*Figure 1a*).

In larvae, IR93a protein is expressed in several neurons in the dorsal organ ganglion, one of the main sensory organs in the larval head (*Stocker, 1994*) (*Figure 1b–c*). These neurons encompass the DOCCs (labeled by an *Ir21a promoter-Gal4*-driven GFP reporter), and the protein localizes prominently to the dendritic bulb at the tip of the sensory processes of these cells (*Figure 1c*). All expression was absent in *Ir93a* mutants, confirming antiserum specificity (*Figure 1c*).

These observations indicated that IR93a might function in cool sensing. Indeed, when larval thermotaxis was assessed on a thermal gradient (*Klein et al., 2015*), we found both *Ir93a* mutant alleles exhibited strong defects in cool avoidance (*Figure 1d*). Cell-specific expression of an *Ir93a* cDNA in the DOCCs under *Ir21a-Gal4* control fully rescued this mutant phenotype (*Figure 1d*). These data demonstrate an essential role for IR93a in DOCCs in larval thermotaxis.

### IR93a is required, together with IR21a and IR25a, for cool-dependent physiological responses of DOCCs

We next assessed whether IR93a is required for the physiological responses of DOCCs to cooling by optical imaging of these neurons using the genetically encoded calcium indicator, GCaMP6m (*Chen et al., 2013*). As previously reported (*Klein et al., 2015*; *Ni et al., 2016*), wild-type DOCCs exhibit robust increases in intracellular calcium in response to cooling (*Figure 2a*). These responses were dramatically reduced in *Ir93a* mutants, and could be rescued by cell-specific expression of an *Ir93a* cDNA (using the *R11F02-Gal4* DOCC driver [*Klein et al., 2015*]) (*Figure 2a,b*). This dramatic loss of DOCC temperature sensitivity resembles that observed in both *Ir21a* and *Ir25a* mutants (*Ni et al., 2016*), and is consistent with IR21a, IR25a and IR93a functioning together to mediate cool activation of the DOCCs.

To provide a more direct readout of thermotransduction in these neurons than soma calcium measurements, we tested the requirement for IR93a, IR21a and IR25a in cool-evoked membrane voltage changes using the genetically encoded voltage sensor, Arclight (*Jin et al., 2012*). In wild-type animals, cool-dependent voltage changes were observed in the DOCC sensory dendritic bulbs (*Figure 2c–d*), where IRs are localized (*Figure 1c*). This response was completely eliminated in *Ir21a*, *Ir25a* and *Ir93a* mutants (*Figure 2c–e*), indicating that each of these IRs is required for temperature-dependent voltage changes in this sensory compartment.

### IR93a is co-expressed with IR25a and IR40a in the antennal sacculus

In adults, *Ir93a* transcripts were previously weakly detected in a set of neurons in the third antennal segment surrounding the sacculus, a three-chambered pouch whose opening lies on the posterior surface of the antenna (*Benton et al., 2009*) (*Figure 3a*). With our IR93a antibody, we detected IR93a expression in neurons innervating sacculus chamber I (11.0 ± 0.5 neurons, n = 48 animals; mean ± SEM) and chamber II (13.9 ± 0.7 neurons, n = 23), with signal detected both in the soma and in the sensory cilia that project into cuticular sensory hairs (sensilla) (*Figure 3b*). As in larval DOCCs,



**Figure 1.** IR93a is expressed in Dorsal Organ Cool Cells (DOCCs) and is required for cool avoidance. (**a**) Gene structure of the *Ir93a* locus; sequences encoding the transmembrane (TM) domains and channel pore are colored. The blue triangle denotes site of *MiMIC* insertion in *Ir93a^{MI05555}*, and the CRISPR/Cas9-generated deletion in the *Ir93a^{122}* allele is shown below. (**b**) Schematic of the larval anterior showing the bilaterally symmetric Dorsal Organ Ganglia (grey) within which three Dorsal Organ Cool Cells (DOCCs) are located. (**c**) Immunofluorescence of the larval anterior (corresponding to the boxed region in the schematic) showing expression of IR93a protein (magenta) in DOCCs (*Ir21a-Gal4;UAS-GFP* [*Ir21a>GFP*]) (green), as well as additional sensory neurons. *Ir93a^{MI05555}* mutants lack IR93a immunostaining. The arrow and arrowhead label the soma and dendritic bulb of one of the DOCCs. Scale bar is 10 μm. (**d**) Cool avoidance behavior assessed as navigational bias (movement toward warmth / total path length) of individual larval trajectories on an ~0.36°C/cm gradient extending from ~13.5°C to ~21.5°C, with a midpoint of ~17.5°C. Letters denote statistically distinct categories (alpha = 0.05; Tukey HSD). *wild type (Canton-S)*, n = 37 animals. *Ir93a^{MI05555}*, n = 132. *Ir21a-Gal4/+; Ir93a^{MI05555}*, n = 72. *Ir93a^{MI05555},UAS-Ir93a/ Ir93a^{MI05555}*, n = 80. *Ir21a-Gal4/+; Ir93a^{MI05555},UAS-Ir93a/ Ir93a^{MI05555}*, n = 45. *Ir93a^{122}*, n = 101.

IR25a was expressed in IR93a-expressing cells in the sacculus (*Figure 3c*). By contrast, no expression was detected in these cells when using our *Ir21a* promoter driver (data not shown). We found instead that the IR93a/IR25a sacculus neurons express a distinct receptor, IR40a (*Figure 3d–e*) (*Benton et al., 2009*; *Silbering et al., 2016*).



**Figure 2.** Cool-responsive calcium and voltage changes in DOCCs require IR93a. (a) Left: DOCC responses monitored using *R11F02>GCaMP6m*. DOCC cool-responsive increases in fluorescence are dramatically reduced in *Ir93a^MI05555^*, and responses are rescued by expression of a wild-type *Ir93a* cDNA under *R11F02-Gal4* control. Traces, average ± SEM. Right: Ratio of fluorescence at 14°C versus 20°C depicted using a violin plot (internal white circles show median; black boxes denote 25th to 75th percentiles; whiskers extend 1.5 times interquartile range). Letters denote statistically distinct

*Figure 2 continued on next page*

*Figure 2 continued*

categories, p<0.01, Steel-Dwass test. *wild type*, n = 12 cells. *Ir93a*$^{MI05555}$, n = 44. *Ir93a*$^{MI05555}$; *R11F02>Ir93a*, n = 46. (**b**) Temperature-dependent DOCC voltage responses in the sensory endings of *wild-type* (upper panels) or *Ir93a*$^{MI05555}$ mutant (lower panels) larvae monitored using *R11F02>Arclight*. Arrowheads denote DOCC dendritic bulbs. Note that Arclight fluorescence decreases upon depolarization. Asterisks denote cuticular autofluorescence from adjacent sensory structures. (**c**) Robust cool-responsive depolarization of DOCC sensory endings is observed in otherwise *wild-type* animals using either *R11F02>Arclight* or *Ir21a>Arclight*. Depolarization response is eliminated in *Ir93a*$^{MI05555}$, *Ir25a*$^2$, and *Ir21a*$^{\Delta1}$ mutants. Traces, average ± SEM. Violin plot depicts ratio of fluorescence at 14°C versus 20°C. ** denotes distinct from wild-type control, p<0.01 compared to control, Steel-Dwass test. *R11F02-Gal4;UAS-Arclight*, n = 57 cells. *R11F02-Gal4;UAS-Arclight;Ir93a*$^{MI05555}$, n = 24. *R11F02-Gal4;UAS-Arclight; Ir25a*$^2$, n = 30. *Ir21a-Gal4;UAS-Arclight*, n = 18. *Ir21a-Gal4;UAS-Arclight; Ir21a*$^{\Delta1}$, n = 23.

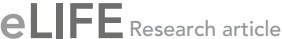

**Figure 3.** IR93a is co-expressed with IR25a and IR40a in sacculus neurons. (**a**) Left: schematic of the adult *Drosophila* antenna, illustrating the location of the sacculus (red) in the interior of this appendage. Right: the sacculus is composed of three main chambers (I, II, III), which are lined with sensilla of various morphologies (cartoon adapted from [*Shanbhag et al., 1995*]). (**b**) Top: immunofluorescence on a whole-mount wild-type antenna showing expression of IR93a protein (green) in two groups of soma (arrows) around sacculus chambers I and II; these chambers are visualized by cuticle autofluorescence shown in the images on the right. The arrowhead marks the concentration of IR93a in the dendritic endings that innervate the sensilla in chamber I. Note that the dendrites of chamber II neurons are not visible in this image; sensilla localization of IR93a in these cells is more easily detected in antennal sections; see panel (**d**). Bottom: *Ir93a*$^{MI05555}$ mutants lack detectable IR93a protein. Scale bar is 20 µm. (**c–e**) Double immunofluorescence with the indicated antibodies on antennal cryosections revealing co-expression of these IRs in sacculus neurons; the arrows point to the cluster of neurons innervating chamber II. Scale bar is 10 µm. IR25a is expressed in additional neurons that do not express IR93a or IR40a because of IR25a's broader role as an olfactory IR co-receptor (*Abuin et al., 2011*).

## IR93a, IR25a and IR40a are necessary for hygrosensory behavior

Morphological studies have suggested that neurons in sacculus chambers I and II are hygroreceptive (*Shanbhag et al., 1995*), raising the possibility that IR93a, IR25a, and IR40a are required for hygrosensory behavior. To test this hypothesis, we adapted an experimental paradigm (*Perttunen and Salmi, 1956*) in which flies choose between regions of differing humidity generated by two underlying chambers: one containing deionized water and the other containing water saturated with a non-volatile solute (ammonium nitrate) to lower its vapor pressure (*Figure 4a*). This assay design created a humidity gradient of ~96% relative humidity (RH) to ~67% RH, with negligible variation in temperature (*Figure 4b*). Consistent with previous observations (*Perttunen and Salmi, 1956*), wild-type flies exhibited a strong preference for lower humidity (*Figure 4c*). This preference was completely eliminated in *Ir93a* and *Ir25a* mutant flies, and significantly reduced, but not abolished, in *Ir40a* mutants (*Figure 4c*, *Figure 4—figure supplement 1a*). All of these behavioral defects were robustly rescued by the corresponding cDNAs, confirming the specificity of the mutant defects (*Figure 4c*). Importantly, the loss of *Ir21a* (or other antennal-expressed IR co-receptors, *Ir8a* and *Ir76b*) did not disrupt dry preference. To exclude any potential contribution of the non-volatile solute to the behavior observed, we also tested flies in a humidity gradient (~89% to ~96% RH) generated using underlying chambers of deionized water alone and air (*Figure 4a*). Even in this very shallow gradient, wild type flies displayed a strong preference for the side with lower humidity (*Figure 4d*), and this preference was dependent on IR93a, IR25a and IR40a, but independent of IR21a (*Figure 4d*). The distinction between the functions of IR21a and IR40a extended to thermotaxis, as *Ir40a* mutants exhibited no defects in this IR21a-dependent behavior (*Figure 4—figure supplement 1a–b*), consistent with lack of expression of IR40a in the larval DOCCs (data not shown).

## IRs mediate dry detection by sacculus neurons

To test whether the IR40a/IR93a/IR25a-expressing sacculus neurons are physiological hygrosensors, we monitored their calcium responses to changes in the RH of an airstream (of constant temperature) directed towards the antenna. We used *Ir40a-Gal4* to express *UAS-GCaMP6m* selectively in these neurons, and measured GCaMP6m fluorescence in their axon termini, which innervate two regions of the antennal lobe, the 'arm' and the 'column' (*Silbering et al., 2016*; *Silbering et al., 2011*) (*Figure 5a–b*).

We observed that these sacculus neurons behave as dry-activated hygrosensors: decreasing the RH from ~90% to ~7% RH elicited an increase in GCaMP6m fluorescence, while increasing RH from ~7% to ~90% elicited a decrease (*Figure 5c–g*). Calcium changes were most apparent in the 'arm' (*Figure 5c*). Importantly, these physiological responses were IR-dependent: mutations in either *Ir93a* or *Ir40a* eliminated the dry response (*Ir25a* mutants were not tested), and these defects were restored with corresponding cDNA rescue transgenes (*Figure 5d–g*). These data corroborate the requirement for IRs in behavioral preference for lower humidity.

## The TRP channels Nanchung and Water witch do not mediate IR-dependent dry sensation

Previous work has implicated two Transient Receptor Potential (TRP) channels, Nanchung and Water witch, in hygrosensation (*Liu et al., 2007*), but it is unclear whether they have an essential function in this modality (*Enjin et al., 2016*; *Ji and Zhu, 2015*) and the cells in which these proteins act are unknown (*Jourjine et al., 2016*; *Liu et al., 2007*). In our gradient assay, we found that animals mutant for *nanchung* or *water witch* displayed partially diminished dry preference behavior (*Figure 6a*). However, neither *nan* nor *wtrw* was required for the dry responsiveness of IR40a-expressing sacculus neurons (*Figure 6b–e*). Thus, these TRP channels are not essential for IR-dependent dry sensing, suggesting that they contribute to hygrotaxis through other mechanisms.

## Discussion

From their ancestral origins within the synaptic iGluR family, IRs are widely appreciated to have evolved functionally diverse roles in environmental chemosensory detection (*Croset et al., 2010*; *Rytz et al., 2013*). Here we provide evidence that a previously uncharacterized member of this repertoire, IR93a, functions in two critical non-chemosensory modalities, thermosensation and

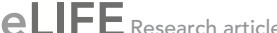

**Figure 4.** Hygrosensory behavior requires IR93a, IR25a and IR40a. (a) Schematic of the hygrosensory behavior assays. ~67% to ~96% RH gradients were generated by filling wells with either a saturated solution of ammonium nitrate in water or pure water. ~89% to ~96% RH gradients were generated by pairing empty wells with wells filled with pure water. Nylon mesh prevented fly contact with solutions. Dry preference was quantified by counting flies on either side of chamber midline. 25–35 flies were used per assay. (b) Mean ± SD of RH and temperature measured at indicated gradient positions. ~67% to ~96% RH, n = 58 gradients. ~89% to ~96% RH, n = 28. (c,d) Dry preference assessed on ~67% vs. ~96% (c) and ~89% vs. ~96% (d) gradients. Asterisks denote statistically distinct from *wild type* (**p<0.01; *p<0.05, Steel with control). *wild type*, n = 16 assays. *Ir8a* mutant (*Ir8a¹*), n = 8. *Ir76b* mutant (*Ir76b²*), n = 14. *Ir21a* mutant (*Ir21a¹²³*), n = 14. *Ir25a* mutant (*Ir26a²*), n = 11. Ir25a rescue (*Ir25a²; UAS-Ir25a*), n = 15. *Ir40a* mutant (*Ir40a¹*), n = 15. *Ir40a* rescue (*Ir40a¹; UAS-Ir40a*), n = 9. *Ir40a* CRISPR mutant (*Ir40a¹³⁴*), n = 10. *Ir93a* mutant (*Ir93a^MI05555*), n = 11. *Ir93a* rescue (*Ir93a^MI05555, UAS-Ir93a*), n = 14. *Ir40a* mutant alleles and thermosensory behavior are shown in *Figure 4—figure supplement 1a–b*. Note that *UAS-cDNA* rescues were observed in the absence of *Gal4* drivers, reflecting *Gal4*-independent expression of *UAS* transgenes (*Figure 4—figure supplement 1c–d*).

The following figure supplement is available for figure 4:

**Figure supplement 1.** Description of Ir40a mutants and analysis of Gal4-independent transgene expression.



**Figure 5.** IR-dependent physiological responses to dry air. (a) Schematic of the *Drosophila* head (viewed from above) illustrating the projection of IR40a/IR93a/IR25a-expressing neurons (green) (labeled using *Ir40a-Gal4* [*Silbering et al., 2011*]) from the sacculus to the antennal lobes in the brain, visualized through a hole in the head cuticle. (b) Raw fluorescence image of *Ir40a* axons (in *Ir40a-Gal4;UAS-GCaMP6m* animals) innervating the arm and column in the antennal lobe. The dashed circle indicates the position of the ROI used for quantification in panels (d–g). (c) Color-coded images (reflecting GCaMP6m fluorescence intensity changes) of IR40a neuron responses to a switch from 90% to 7% RH ('Dry response') and to a switch from 7% to 90% RH ('Moist response'). (d,f) Moisture-responsive fluorescence changes in the arm (moist = 90% RH, dry = 7% RH). Traces represent average ± SEM. (e,g) Quantification of changes in ΔF/F (mean fluorescence change in the ROI shown in [b]) upon shift from moist to dry (e) or dry to moist (g). Dry responses were quantified as [ΔF/F at 7% RH (average from 4.5 to 6.5 s after shift to 7% RH)] - [ΔF/F at 90% RH (average from 3.5 to 1 s prior to shift to 7% RH)], and moist responses quantified by performing the converse calculation. Genotypes: control: n = 17 animals (pooled data from *Ir40a-Gal4, Ir40a$^1$/IR40a-Gal4,+;UAS-GCaMP6m/+*, n = 9; *IR40a-Gal4;UAS-GCaMP6m,Ir93a$^{MI05555}$/+*, n = 8). *Ir93a* mutant (*Ir40a-Gal4;UAS-GCaMP6m,Ir93a$^{MI05555}$/Ir93a$^{MI05555}$*), n = 10. *Ir93a* rescue (*Ir40a-Gal4;UAS-GCaMP6m,Ir93a$^{MI05555}$/UAS-mcherry:Ir93a,Ir93a$^{MI05555}$*), n = 8. *Ir40a* mutant (*Ir40a-Gal4,Ir40a$^1$;UAS-GCaMP6m/+*), n = 8. *Ir40a* rescue (*Ir40a-Gal4,Ir40a$^1$;UAS-GCaMP6m/UAS-Ir40a*), n = 6. **p<0.01, distinct from controls and rescues, Steel-Dwass test.

hygrosensation. In both of these roles, IR93a acts with the broadly expressed co-receptor IR25a. However, these IRs mediate these two modalities in different populations of neurons in conjunction with a third, distinct IR: with IR21a in cool sensation, but not dry sensation, and with IR40a in dry sensation, but not cool sensation. All of these receptors are widely conserved in insects, indicating that these sensory pathways likely underlie behavioral responses of diverse species to these important environmental stimuli.

The identification of an IR21a/IR25a/IR93a-dependent cool-sensing system provides a molecular counterpart to the well-established TRP channel and 'Gustatory' Receptor GR28B(D) warmth-sensing systems (*Barbagallo and Garrity, 2015*; *Ni et al., 2013*). By contrast, despite the importance of hygrosensation in helping insects to avoid desiccation or inundation (*Chown et al., 2011*) and – in



**Figure 6.** The TRP channels Nanchung and Water witch do not mediate IR-dependent dry sensation. (a) Dry preference assessed on ~67% to ~96% gradient. Asterisks denote statistically different responses from *wild type* (**p<0.01; Steel with control). *wild type*, n = 16 assays. *nan* mutant (*nan³⁶ᵃ*), n = 9. *wtrw* mutant (*wtrw²*), n = 9. (b–e) Moisture-responsive fluorescence changes of IR40a neurons recorded and quantified as described in *Figure 5*. Genotypes: control: n = 5 animals (*Ir40a-Gal4,UAS-GCaMP6m/+*). *nan* mutant (*Ir40a-Gal4,UAS-GCaMP6m/+;nan³⁶ᵃ*), n = 5. *wtrw* mutant (*Ir40a-Gal4, UAS-GCaMP6m/+ ;wtrw²*), n = 7. (All P>0.4 versus control, Steel with control).

blood-feeding species such as mosquitoes – to locate mammalian hosts (*Brown, 1966*; *Olanga et al., 2010*), the neuronal and molecular basis of this sensory modality is poorly understood. Hygrosensitive neurons have been identified electrophysiologically in large insects (*Tichy and Gingl, 2001*; *Tichy and Kallina, 2010*), but their behavioral role has been hard to determine. In *Drosophila*, the antenna has long been suspected to be an important hygrosensory organ (*Perttunen and Syrjamaki, 1958*; *Sayeed and Benzer, 1996*), but there has been little consensus on the relevant populations of neurons and sensory receptors (*Ji and Zhu, 2015*; *Liu et al., 2007*;

*Yao et al., 2005*). We have identified a discrete population of dry-activated hygroreceptors in the sacculus that express IR40a/IR93a/IR25a. Together with an independent study (*Enjin et al., 2016*), our data provide physiological and behavioral evidence supporting these as one pathway that enables flies to distinguish external humidity levels.

In addition to the roles of IR93a in cool and dry sensing, it is very likely that this receptor defines additional sensory pathways. Our expression analysis has identified IR93a-positive cells that do not express IR21a or IR40a, such as non-DOCCs in the larval dorsal organ (*Figure 1c*). Moreover, the milder hygrosensory behavior phenotype of our protein null *Ir40a* mutants compared to *Ir93a* (or *Ir25a)* mutants hints that IR93a may have broader roles in this sensory modality than acting exclusively with IR40a. The populations of IR93a-expressing neurons characterized in this study are themselves heterogeneous. For example, IR40a/IR93a/IR25a-expressing sacculus neurons belong to two morphologically and physiologically distinct subpopulations. Arm neurons have contralateral projections (*Silbering et al., 2011*) and respond robustly to low humidity, while column neurons are exclusively ipsilateral (*Silbering et al., 2011*) and respond more weakly to humidity, as well as displaying mild thermosensitivity (*Enjin et al., 2016*). IR40a-expressing neurons also respond to ammonia (*Silbering et al., 2016*). Given that these neurons are housed in apparently poreless sensilla (*Shanbhag et al., 1995*), we speculate that this chemical activates these cells indirectly, for example, through modification of the humidity of the air, or the temperature of the cuticular surface, within the sacculus.

A key future challenge will be to determine the mechanisms by which IRs contribute to the sensation of thermal and humidity cues. We previously showed that ectopically-expressed IR21a can confer cool sensitivity to other IR-expressing neurons, consistent with IR21a acting as a sensory specificity determinant (*Ni et al., 2016*). It will be important to determine if IR40a serves a more permissive role or functions in a similar capacity in dry sensing. The contribution (if any) of the Venus flytrap-like ligand-binding domains of these receptors is of particular interest. Although this domain recognizes glutamate in iGluRs, and diverse organic molecules in chemosensory IRs, it is conceivable that these domains mediate thermo- and hygrosensory detection in these receptors in a ligand-independent manner. For example, IR21a could transduce information via temperature-dependent conformational changes. The requirement for IR93a (and IR25a) in both thermosensation and hygrosensation also indicates that these modalities could share common mechanisms of sensory detection. For example, hygrosensation could involve a thermosensory component, based on evaporative cooling. Alternatively, both temperature and moisture detection could involve mechanosensation, based on swelling or shrinkage of sensory structures, as suggested in mammals and *C. elegans* (*Filingeri, 2015*; *Russell et al., 2014*). Further characterization of how IRs mediate temperature and moisture detection is currently limited by our inability to reconstitute thermosensory or hygrosensory responses in heterologous systems by expressing the known combinations of IRs (G.B., L.N., A.F.S., R.B. and P.G., unpublished data). IRs, like iGluRs, are thought to form heterotetrameric complexes (*Abuin et al., 2011*), raising the possibility that additional IR subunits are required. It is also conceivable that other types of accessory signaling molecules act with IRs, and/or that the cellular and cuticular specializations of the thermosensory and hygrosensory structures are critical to allow monitoring of these ubiquitous and ever-changing environmental stimuli.

## Materials and methods

### Fly strains

*Ir25a²* (*Benton et al., 2009*), UAS-Ir25a (*Abuin et al., 2011*), *Ir8a¹* (*Abuin et al., 2011*), *Ir21a¹²³* (*Ni et al., 2016*), *Ir76b²* (*Zhang et al., 2013*), R11F02-Gal4 (*Klein et al., 2015*), Ir40a-Gal4 (*Silbering et al., 2011*), *Ir40a¹* (*Silbering et al., 2016*), UAS-Ir40a (*Silbering et al., 2016*), *Ir93aᴹᴵ⁰⁵⁵⁵⁵* (*Venken et al., 2011*), UAS-GCaMP6m (P[20XUAS-IVS-GCaMP6m]attp2 and P[20XUAS-IVS-GCaMP6m]attp2attP40 [*Chen et al., 2013*]), UAS-Arclight (*Cao et al., 2013,*) UAS-GFP (P[10XUAS-IVS-Syn21-GFP-p10]attP2 [*Pfeiffer et al., 2012*]), *nan³⁶ᵃ* (*Gong et al., 2004*), *wtrw²* (*Kwon et al., 2010*) and *y¹ P(act5c-cas9, w) M(3xP3-RFP.attP)ZH-2A w** (*Port et al., 2014*) were previously described.

*Ir40a¹³⁴* (*Figure 4—figure supplement 1a*) and *Ir93a¹²²* (*Figure 1a*) were generated by transgene-based CRISPR/Cas9-mediated genome engineering (*Port et al., 2014*), using either an *Ir40a-*

targeting gRNA (5'-GCCCGTTTAAGCAAGACATC) or an *Ir93a*-targeting gRNA (5'-TCAGCAGAA TGATGCCCATT) expressed under U6-3 promoter control (dU6-3:gRNA) in the presence of *act-cas9*. *UAS-mCherry:Ir93a* contains codons 29–869 of the *Ir93a* ORF (corresponding to *Ir93a-PD* [flybase. org], without the sequence encoding the predicted endogenous signal peptide), which were PCR amplified from Oregon R antennal cDNA and subcloned into *pUAST-mCherry attB* (*Abuin et al., 2011*) (which encodes the calreticulin signal sequence upstream of the *mCherry* ORF). This construct was integrated into VK00027 by phiC31-mediated transgenesis (Genetic Services, Inc.).

## Behavior

Thermotaxis of early second instar larvae was assessed over a 15 min period on a temperature gradient extending from 13.5 to 21.5°C over 22 cm (~0.36°C/cm) as described (*Klein et al., 2015*). As thermotaxis data were normally distributed (as assessed by Shapiro-Wilk test), statistical comparisons were performed by Tukey HSD test, which corrects for multiple comparisons.

To assay hygrosensory behavior, 8 well rectangular dishes (12.8 × 8.55 × 1.5 cm; ThermoFisher #267060) were modified to serve as humidity preference chambers. The lids of two 8 well plates were used. A heated razor blade was used to cut out the middle of one lid, and a nylon mesh was glued into place around the edges, providing a surface for the animals to walk on which separated them from contact with any liquid. A soldering iron was used to melt a small hole in a second culture plate lid, which could then be placed over the screen, creating a chamber ~0.7 cm in height in which the flies could move freely. To monitor the gradients formed, an additional chamber was constructed with four holes equally spaced along its length to allow the insertion of humidity sensors (Sensirion EK-H4 evaluation kit) for monitoring the humidity and temperature.

Prior to the start of each experiment, 4 wells on one side of the culture dish were filled with purified water, while the opposite 4 were filled with ~4 ml water and sufficient ammonium nitrate to obtain a saturated solution (~3 g). The gradient was assembled with the screen and lid piece, and the whole apparatus wrapped in food service film to avoid any transfer of air between the inside and outside of the device. Gradients were transferred to an environmental room that maintained at constant external temperature and humidity (25°C and 70%RH). Ammonium nitrate gradients were permitted to equilibrate for approximately 1 hr and were stable over many hours. For the water and air only gradients, the air only side humidified over time. These gradients were incubated for 25 min prior to use to allow the temperature to equilibrate; the humidity of the dry side typically rose by ~2% RH during the 30 min assay (values shown are at the 30 min time point). A small hole was poked through the food service film covering the device to allow animals to be transferred to the gradient. This hole was sealed using transparent scotch tape once the animals were inside. Experiments used 1–4 day old adult flies that had been sorted under light CO2 anesthesia into groups of 30 (15 male and 15 female) animals 24 hr before testing, and transferred to fresh tubes. Flies were allowed 30 min to settle on the gradient, at which point a photograph was taken of their position, and the number of animals on each side counted, allowing calculation of a dry preference index as follows:

$$\text{Dry Preference} = \frac{\#\ \text{animals on dry side} - \#\ \text{animals on moist side}}{\text{total}\ \#\ \text{of animals}}$$

As moisture preference data did not conform to normal distributions (as assessed by Shapiro-Wilk test, p<0.01), statistical comparisons to wild-type control were performed by Steel test, a nonparametric test that corrects for multiple comparisons, using JMP11 (SAS).

## Calcium and Arclight imaging

Calcium and Arclight imaging of larval thermosensors was performed as previously described (*Klein et al., 2015*). Pseudocolor images were created using the 16-colors lookup table in ImageJ 1.43r. Adult antennal lobe calcium imaging was performed as described for olfactory imaging (*Silbering et al., 2012*), with slight modifications to sample preparation and stimulation. Briefly, 3–7 day old flies were fixed to a Plexiglas stage using UV-glue (A1 Tetric Evoflow, Ivoclar Vivadent), the antennae were pulled forward and a small opening was made in the head capsule to allow visual access to the antennal lobes. For the stimulation compressed air from a tank was passed through activated charcoal and then either through an empty gas washing bottle or a gas washing bottle filled with distilled water producing either a dry airstream of ~7% RH or a humid airstream of

~90% RH. A computer controlled solenoid valve (The Lee Company, Westbrook, CT) was used to switch the airflow between the two gas washing bottles. The flow was kept constant at 1 l/min with a parallel arrangement of two 500 ml/min mass flow controllers (PKM SA, www.pkmsa.ch) placed before the gas washing bottles. Activating the solenoid valve resulted in a complete reversal of RH from low to high or high to low within less than 10 s. For each animal tested, both high to low and low to high RH transitions were applied in random order. Following humidity stimulation, a final pulse of 10% ammonia was applied as a control to confirm cellular activity (*Silbering et al., 2016*) (animals showing no response to this positive control were excluded from the analysis). Data were processed using Stackreg (ImageJ) (*Thevenaz et al., 1998*) to correct for movement artifacts (animals with movement artifacts that could not be corrected with Stackreg were excluded from the analysis) and custom scripts in Matlab and R as previously described (*Silbering et al., 2011*). As quantified imaging data did not conform to normal distributions (as assessed by Shapiro-Wilk test, $p < 0.01$), statistical comparisons were performed by Steel-Dwass test, a non-parametric test that corrects for multiple comparisons, using JMP11 (SAS).

## Immunohistochemistry

Larval immunostaining was performed as described (*Kang et al., 2012*). Immunofluorescence on antennal cryosections or whole-mount antennae was performed essentially as described (*Saina and Benton, 2013*), except that whole-mount antennae were placed in Vectashield immediately after the final washes without dehydration. The following antibodies were used: rabbit anti-IR25a (1:1000; [*Benton et al., 2009*]), guinea pig anti-IR40a (1:200, [*Silbering et al., 2016*]), rabbit anti-IR93a (peptide immunogen CGEFWYRRFRASRKRRQFTN, Proteintech, Rosemont, IL, USA, 1:4000 for tissue sections and 1:500 for whole-mount tissue), guinea pig anti-IR25a (peptide immunogen SKAALRPRF NQYPATFKPRF, Proteintech, Rosemont, IL, USA, 1:200), mouse anti-GFP (1:200; Roche), goat anti-rabbit Cy3 (1:100 larva, 1:1000 sections; Jackson ImmunoResearch), goat anti-rabbit Alexa488 (1:100 antenna whole-mount, 1:1000 antennal sections, A11034 Invitrogen AG), goat anti-guinea pig (1:1000, A11073 Invitrogen AG) and donkey anti-mouse FITC (1:100; Jackson ImmunoResearch).

## RT-PCR

cDNA for each genotype was purified from 20 fly heads (RETROscript, Ambion) for RT-PCR. Primers used: *Ir25a* forward, 5'-TAGCAGTCAGCGGGACAATG; *Ir25a* reverse, 5' -GAGTGGATTGCGTGAC-GAGA; *Ir40a* forward, 5'-GGCGAGGACAAGGCAGTA; *Ir40a* reverse, 5'-CGGCAGCGGTCATCTTA TCT; *Ir93a* forward, 5'-TGCCAAGGTCCAGCAGATTC; *Ir93a* reverse, 5'-AACATGTTCAGGGTC TCGGC. *RpL32* forward, 5'-GCTAAGCTGTCGCACAAATG; RpL32 reverse 5'-GTTCGATCCG TAACCCGATGT.

## Acknowledgements

This work was supported by a grant from the National Institute on Deafness and Other Communication Disorders (F31 DC015155) to ZAK, the National Institute of Neurological Disorders and Stroke (F32 NS077835) to MK, a Boehringer Ingelheim Fonds PhD Fellowship to R Bell, European Research Council Starting Independent Researcher and Consolidator Grants (205202 and 615094) and a Swiss National Science Foundation Project Grant (31003A_140869) to R Benton, the National Institute of General Medical Sciences (F32 GM113318) to GB, the National Institute of Allergy and Infectious Diseases (R01 AI122802) to PAG, and the National Institute of General Medical Sciences (P01 GM103770) to ADTS and PAG.

## Additional information

### Funding

| Funder | Grant reference number | Author |
| --- | --- | --- |
| National Institutes of Health | F31 DC015155 01A1 | Zachary A Knecht |
| National Institutes of Health | F32 NS077835 | Mason Klein |
| National Institutes of Health | F32 GM113318 | Gonzalo Budelli |

| Boehringer Ingelheim Fonds | PhD Fellowship | Rati Bell |
| National Institutes of Health | P01 GM103770 | Paul A Garrity<br>Aravinthan DT Samuel |
| European Research Council | Starting Independent Researcher Grant 205202 | Richard Benton |
| European Research Council | Consolidator Grant 615094 | Richard Benton |
| Schweizerischer Nationalfonds zur Förderung der Wissenschaftlichen Forschung | 31003A_140869 | Richard Benton |
| National Institutes of Health | R01 AI22802 | Paul A Garrity |

The funders had no role in study design, data collection and interpretation, or the decision to submit the work for publication.

## Author contributions

ZAK, Performed molecular genetics, behavior, immunohistochemistry, neurophysiology and data analysis, Conception and design, Drafting or revising the article; AFS, Performed neurophysiology, immunohistochemistry and data analysis, Conception and design, Drafting or revising the article; LN, Performed molecular genetics, neurophysiology, behavior, immunohistochemistry and data analysis, Conception and design, Drafting or revising the article; MK, Performed neurophysiology, behavior and data analysis, Conception and design, Drafting or revising the article; GB, Performed neurophysiology, Drafting or revising the article; RBel, Performed molecular genetics, Drafting or revising the article; LA, Performed immunohistochemistry, Acquisition of data, Drafting or revising the article; AJF, Analysis of data, Drafting or revising the article; ADTS, RBen, PAG, Conception and design, Analysis and interpretation of data, Drafting or revising the article

## Author ORCIDs

Paul A Garrity, http://orcid.org/0000-0002-8274-6564

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
