## [Decision Letter]

Thank you for submitting your article "Distinct combinations of variant ionotropic glutamate receptors mediate thermosensation and hygrosensation in *Drosophila*" for consideration by *eLife*. Your article has been favorably evaluated by K VijayRaghavan (Senior Editor) and three reviewers, one of whom, Mani Ramaswami (Reviewer #1), is a member of our Board of Reviewing Editors. The following individual involved in review of your submission has agreed to reveal their identity: Hubert Amrein (Reviewer #2).

The reviewers have discussed the reviews with one another and the Reviewing Editor has drafted this decision to help you prepare a revised submission.

The elucidation of the protein composition of different non-chemosensory Ir channel complexes is relevant to numerous fields of neurobiology. Knecht and colleagues show in this paper that Ir proteins play essential roles in cool sensing of larvae and humidity sensing of adult flies. Ir proteins have been largely studied in context of chemosensory perception, albeit they were recently invoked in temperature and humidity sensing. The main conclusion from the study is that Ir proteins combine into different heteromeric complexes to mediate different environmental signals. Chemosensory neurons expressing different combinations of IRs have been shown to detect diverse chemicals. In this study, Knecht and colleagues report that dorsal organ sensory neurons in *Drosophila* larvae expressing IR25a, IR93a and IR21a respond to cooling temperature, whereas olfactory sensory neurons in adult antennae expressing IR25a, IR93a and IR40a respond to dry air. Despite the recent publication of an independent study that reaches many of the same conclusions regarding humidity sensing (Enjin et al., 2016), numerous neat genetic reagents were generated in this study, with which the authors have an opportunity to advance the field. The study is of high quality. All experiments are well controlled, and the conclusions are sound and supported by the presented data. The paper is also well-written.

Essential revisions:

1) Apparently, simply replacing IR21a by IR40a in a combination of three IRs changes the modality of the sensory neurons from temperature to humidity. This is very interesting. However, the paper should be revised to include strong and specific molecular hypotheses for how trimeric combination of IRs confer specificity to humidity or temperature. Do their models predict that simple ectopic expression of Ir40a in DOCCs (in either wild-type or IR21a null backgrounds) will cause these neurons to respond to dry air? Similarly, do their models predict that ectopic expression of IR21a in the saccular neurons (in wild-type or IR40a mutant background) will make them respond robustly to cooling temperature? Or is there reason to believe that other unidentified molecules in the larval or adult neurons and/or specific structural features (as proposed by EM studies) are necessary to specify sensory modality?

This issue should be addressed clearly and, if the authors consider useful, then experimentally. If there is no experimental result to show that any of these IRs plays more than a permissible role, then the text (including the Abstract) should be modified and nuanced to explicitly acknowledge key ambiguities and/or complexities.

2) Silbering et al., Nature 2016 show that Ir40a neurons do not respond to DEET as previously published an observation that triggers a retraction of the DEET paper in the same issue of Nature. However, Figure 1 of Silberling et al. 2016 (which has an overlapping authorship with this manuscript) shows that IR40a neurons in the sacculus respond strongly to ammonia and that this ammonia response is lost in Ir40a mutants. This observation is superficially inconsistent with the current study – which indicates a role for IR40a in humidity sensing. It is also inconsistent with prior anatomical studies (e.g. Shanbhag et al., 1995) that many insect sensilla in the succulus, unlike other olfactory sensilla, do not have pores detectable by electron microscopy, making it unlikely for external chemicals to reach the sensory neurons in the saccular sensilla.

The apparent dichotomy needs to be explicitly reconciled and discussed, for clarity and for the field as one major contribution of this paper is to clarify and define the function of Ir40a. A potential explanation is that ammonia is highly hygroscopic (absorbing water from the air) and therefore may reduce humidity in the local environment surrounding the antennal neurons. In this scenario, the response to ammonia would be secondary to its effect on humidity, explaining why and how IR40a mutations can abolish sensory response to ammonia as well as sensory response to dry air. Experimentally, it would be important to clarify this perhaps simply by asking whether filling a bottle with 10% ammonia would change the humidity in the bottle. And /or whether the ammonia response of these neurons is lost if ammonia is applied to IR40a cells along with saturated water vapour.

---

## [Author Response]

*Essential revisions:*

*1) Apparently, simply replacing IR21a by IR40a in a combination of three IRs changes the modality of the sensory neurons from temperature to humidity. This is very interesting. However, the paper should be revised to include strong and specific molecular hypotheses for how trimeric combination of IRs confer specificity to humidity or temperature. Do their models predict that simple ectopic expression of Ir40a in DOCCs (in either wild-type or IR21a null backgrounds) will cause these neurons to respond to dry air? Similarly, do their models predict that ectopic expression of IR21a in the saccular neurons (in wild-type or IR40a mutant background) will make them respond robustly to cooling temperature? Or is there reason to believe that other unidentified molecules in the larval or adult neurons and/or specific structural features (as proposed by EM studies) are necessary to specify sensory modality?*

*This issue should be addressed clearly and, if the authors consider useful, then experimentally. If there is no experimental result to show that any of these IRs plays more than a permissible role, then the text (including the Abstract) should be modified and nuanced to explicitly acknowledge key ambiguities and/or complexities.*

We have attempted numerous IR21a and IR40a ectopic expression experiments, both in flies and in heterologous cell types. In the case of Ir21a, we recently reported that ectopically expressing IR21a in fly aristal “hot sensing” neurons could confer cool-responsiveness and that this required the presence of the co-receptor IR25a (see Figure 5 and Figure 6 of Ni et al. 2016), demonstrating an instructive role for IR21a in cool sensing. However, our attempts to reconstitute a cold receptor in heterologous cells (including S2 cells, HEK cells and frog oocytes) by simply co-expressing Ir21a, Ir25a and Ir93a have not been successful. For IR40a, since the internally-located DOCCs cannot be exposed to a dry air stream, we misexpressed this receptor in IR-dependent olfactory neurons on the antenna, but this did not confer dry sensitivity. Thus, we have shown an instructive role for IR21a in cool sensing, but so far only a permissive role for IR40a in dry sensing.

While negative results from ectopic expression experiments must be treated with caution, they are consistent with the requirement for additional cell-specific factors to reconstitute the responses. At the molecular level, such factors could include auxiliary proteins that promote IR folding, subunit assembly, transport or activity, and/or signaling molecules that collaborate with the IRs. In addition, the full expression of these modalities may require the morphologically specialized sensory sensilla that house the receptors.

In the revised Abstract, we are careful only to state that we have shown the IRs are important for the responses (rather than “underlying” the responses as previously stated):

“Our results identify IR93a as a common component of molecularly and cellularly distinct IR pathways that are important for thermosensation and hygrosensation in insects.”

We also modified the Discussion to explicitly emphasize the extent of what we do and do not show:

“A key future challenge will be to determine the mechanisms by which IRs contribute to the sensation of thermal and humidity cues. We previously showed that ectopically-expressed IR21a can confer cool sensitivity to IR-expressing neurons, consistent with IR21a acting as a sensory specificity determinant (Ni et al., 2016). It will be important to determine if IR40a serves a more permissive role or functions in a similar capacity in dry sensing.”

*2) Silbering et al., Nature 2016 show that Ir40a neurons do not respond to DEET as previously published an observation that triggers a retraction of the DEET paper in the same issue of Nature. However, Figure 1 of Silberling et al. 2016 (which has an overlapping authorship with this manuscript) shows that IR40a neurons in the sacculus respond strongly to ammonia and that this ammonia response is lost in Ir40a mutants. This observation is superficially inconsistent with the current study – which indicates a role for IR40a in humidity sensing. It is also inconsistent with prior anatomical studies (e.g. Shanbhag et al., 1995) that many insect sensilla in the succulus, unlike other olfactory sensilla, do not have pores detectable by electron microscopy, making it unlikely for external chemicals to reach the sensory neurons in the saccular sensilla.*

*The apparent dichotomy needs to be explicitly reconciled and discussed, for clarity and for the field as one major contribution of this paper is to clarify and define the function of Ir40a. A potential explanation is that ammonia is highly hygroscopic (absorbing water from the air) and therefore may reduce humidity in the local environment surrounding the antennal neurons. In this scenario, the response to ammonia would be secondary to its effect on humidity, explaining why and how IR40a mutations can abolish sensory response to ammonia as well as sensory response to dry air. Experimentally, it would be important to clarify this perhaps simply by asking whether filling a bottle with 10% ammonia would change the humidity in the bottle. And /or whether the ammonia response of these neurons is lost if ammonia is applied to IR40a cells along with saturated water vapour.*

Indeed, EM analyses (Shanbhag et al. 1995) indicate that the Ir40a-Gal4-positive hygrosensory neurons are housed within apparently poreless sensilla that morphologically resemble the hygrosensory sensilla of large insects. This raises the possibility that ammonia activates these neurons not through a classical ligand/chemoreceptor interaction, but rather by altering the properties of the environment that surrounds the sensilla.

As suggested, we attempted to measure the effect of 10% ammonia on relative humidity. We saw no consistent evidence of large humidity or temperature changes; however, we consider this analysis to be unreliable because we found that ammonia vapor permanently damaged our sensors, causing them to give erratic readings. We also measured effects on evaporative cooling, a humidity-related phenomenon that can be measured using a temperature probe resistant to ammonia (Figure 7). We found ammonia can cause a greater cooling effect so in theory could influence sensilla temperature and/or humidity within the sacculus. However, we caution that this remains a speculative scenario as it is difficult to relate what is being measured in the evaporative cooling assay to what occurs in the sacculus.

Author response image 1.Ammonia increases the evaporative cooling effect of water.A temperature probe (Almemo Pt100, ZA9030-FS2) was dipped into water or 3% ammonia and immediately removed.The temperature of the probe was recorded at 1 Hz during 10 s before and 50 s after dipping. The arrowhead indicated the time of dipping into the solution. Traces show temperature changes (mean±SEM; n = 10-11) relative to the mean room temperature before dipping. The drop in temperature reflects evaporative cooling. This phenomenon is more pronounced after dipping the probe in ammonia than in water.**DOI:**
http://dx.doi.org/10.7554/eLife.17879.009

Nevertheless, in an effort to reconcile the current and previous findings, we put forward these possibilities in the Discussion:

“IR40a-expressing neurons also respond to ammonia (Silbering et al., 2016). Given that these neurons are housed in apparently poreless sensilla (Shanbhag et al. 1995), we speculate that this chemical compound activates these cells indirectly, for example, through modification of the humidity of the air or the temperature of the cuticular surface within the sacculus.”